# Update on Pediatric Mild Traumatic Brain Injury in Rural and Underserved Regions: A Global Perspective

**DOI:** 10.3390/jcm12093309

**Published:** 2023-05-06

**Authors:** John K. Yue, Nishanth Krishnan, John P. Andrews, Alexa M. Semonche, Hansen Deng, Alexander A. Aabedi, Albert S. Wang, David J. Caldwell, Christine Park, Melessa Hirschhorn, Kristen T. Ghoussaini, Taemin Oh, Peter P. Sun

**Affiliations:** 1Department of Neurosurgery, Division of Pediatric Neurosurgery, University of California San Francisco, San Francisco, CA 94143, USA; 2Department of Neurosurgery, University of Pittsburgh Medical Center, Pittsburgh, PA 15213, USA; 3Department of Neurosurgery, Duke University, Durham, NC 27708, USA; christine.park@duke.edu; 4Department of Neurosurgery, University of Utah, Salt Lake City, UT 84132, USA

**Keywords:** brain concussion, epidemiology, evidence-based practice, healthcare disparities, medically underserved area, mild traumatic brain injury, pediatric, rural health, socioeconomic factors, telemedicine

## Abstract

Background: Mild traumatic brain injury (MTBI) causes morbidity and disability worldwide. Pediatric patients are uniquely vulnerable due to developmental and psychosocial factors. Reduced healthcare access in rural/underserved communities impair management and outcome. A knowledge update relevant to current gaps in care is critically needed to develop targeted solutions. Methods: The National Library of Medicine PubMed database was queried using comprehensive search terms ((“mild traumatic brain injury” or “concussion”) and (“rural” or “low-income” or “underserved”) and (“pediatric” or “child/children”)) in the title, abstract, and Medical Subject Headings through December 2022. Fifteen articles on rural/underserved pediatric MTBI/concussion not covered in prior reviews were examined and organized into four topical categories: epidemiology, care practices, socioeconomic factors, and telehealth. Results: Incidences are higher for Individuals in rural regions, minorities, and those aged 0–4 years compared to their counterparts, and are increasing over time. Rural healthcare utilization rates generally exceed urban rates, and favor emergency departments (vs. primary care) for initial injury assessment. Management guidelines require customization to resource-constrained settings for implementation and adoption. Decreased community recognition of the seriousness of injury is a consensus challenge to care provision by clinicians. Low parental education and income were correlated with decreased MTBI knowledge and worse outcome. Telehealth protocols for triage/consultation and rehabilitation were feasible in improving care delivery to rural and remote settings. Conclusions: Pediatric MTBI/concussion patients in rural/underserved regions experience increased risks of injury, geographic and financial healthcare barriers, and poorer outcomes. Globally, under-reporting of injury has hindered epidemiological understanding. Ongoing MTBI education should be implemented for rural caregivers, schools, and low-income populations to improve community awareness. Telehealth can improve care delivery across acuity settings, and warrants judicious inclusion in triage and treatment protocols.

## 1. Introduction

Traumatic brain injury (TBI) is a major cause of disability and poses significant strain on care systems worldwide [1]. Children are a uniquely vulnerable population in TBI due to increased risks of delayed developmental milestones, mental health issues, and repetitive TBI [2,3,4]. The estimated global incidence of pediatric TBI is 47–280 per 100,000 and is increasing [1,5,6]. Children aged 0–4 years and young adults aged 15–24 years are at highest risk of hospitalized TBI in developed countries [7,8]. Global costs of pediatric TBI care are difficult to quantify due to lack of published data; in the United States (US) alone, pediatric patients constitute approximately 7% of hospitalized TBI and 15–20% of emergency department (ED) visits [9], with estimated annual costs per patient at USD 12,000 [10]. Costs for hospitalized pediatric TBI easily exceed USD 1 billion per year in the US, excluding additional costs incurred after discharge (e.g., post-acute care and rehabilitation) [11].

Over 80% of pediatric TBI is considered mild (MTBI), commonly defined by an initial Glasgow Coma Scale (GCS) score of 13–15, loss of consciousness under 30 min, and post-traumatic amnesia under 24 h; this spectrum also includes concussions, defined as MTBI with negative computed tomography (CT) findings [12,13,14]. MTBI patients often present to care with mild or subclinical neurologic impairment, and its true incidence is likely higher than published statistics due to under-reporting, inadequate triage to care center, and reduced public awareness, amongst other reasons [15]. While the majority of MTBI/concussion patients are discharged from the ED and do not require neurosurgical intervention, growing evidence linking MTBI with long-term motor, psychosocial, and learning disabilities in children underscores that this disease process should not be regarded as “mild” [16,17,18], and actually warrants dedicated clinical follow-up protocols [19]. Accordingly, analyses have shown that the aggregate costs for pediatric MTBI may exceed that of moderate-to-severe TBI [10].

Mortality after pediatric TBI is associated with lower socioeconomic status (SES), family income, and firearm injuries [1], emphasizing the need to better understand care disparities as risk factors in order to devise strategies for their resolution. In underserved communities, MTBI-related disabilities are often amplified in severity, duration, and worsened trajectories due to challenges in triage, transport, treatment, and rehabilitation [20]. Even in developed nations, rural pediatric patients are at risk for delayed access to high-level care, decreased availability of specialists, increased costs of care, and loss to follow-up [20,21].

Recent topical reviews have focused on the survey of rural MTBI in developed nations [20,22,23]. As international awareness for the importance of MTBI prevention, triage, and acute and post-acute management continue to evolve as part of modern trauma care [9], a knowledge update from the global perspective relevant to the current gaps in care is critically needed to inform the development of targeted solutions. Herein we provide this update through a focused examination of recent pediatric MTBI data across international rural and underserved communities, across four targeted topics of epidemiology, care practices, socioeconomic factors, and telehealth.

## 2. Materials and Methods

The National Library of Medicine PubMed database was queried for primary literature focusing on the study of pediatric MTBI/concussion in rural and medically or socioeconomically underserved regions without limitations by country, and the search term was designed to comprehensively include key words and standard PubMed Medical Subject Headings (MeSH) Terms to achieve this objective: (“mild traumatic brain injury” [title/abstract] *OR* “mild TBI” [title/abstract] *OR* “MTBI” [title/abstract] *OR* “concussion” [title/abstract] *OR* “brain concussion” [MeSH Terms]) *AND* (“rural” [title/abstract] *OR* “rural population” [MeSH Terms] *OR* “rural health” [MeSH Terms] *OR* “rural health services” [MeSH Terms] *OR* “Hospitals, Rural” [MeSH Terms] *OR* “low income” [title/abstract] *OR* “low-income” [title/abstract] *OR* “poverty” [MeSH Terms] *OR* “underserved” [title/abstract] *OR* “medically underserved area” [MeSH Terms]) *AND* (“pediatric” [title/abstract] *OR* “paediatric” [title/abstract] *OR* “pediatrics” [MeSH Terms] *OR* “child” [title/abstract] *OR* “children” [title/abstract] *OR* “child” [MeSH Terms]). No date delimiter was included in the search criteria.

In total, 41 studies from 1991 through December 2022 were identified. Articles were reviewed independently by authors J.K.Y., N.K., and A.M.S. to determine their relevance to modern pediatric MTBI care in rural and/or underserved populations without restriction on nation or locale. As our aim was to advance the knowledge base for global pediatric MTBI care, we selected articles not already covered by topical literature reviews (i.e., [20,22,23,24]) related to rural pediatric MTBI. Studies considering the whole spectrum of pediatric TBI, or those where granular information about MTBI/concussion could not be retrieved, were excluded from analysis. Articles were exclude based on the following criteria: article covered by recent topical review (N = 6), lack of rural focus (N = 5), lack of MTBI focus (N = 5), review article (N = 4), conference abstract (N = 2), dental trauma article (N = 2), and opinion article (N = 2) (Figure 1).

Fifteen articles were selected unanimously for inclusion due to their relevance and were stratified by topic: epidemiology (N = 3), care practices (N = 5), socioeconomic factors (N = 5), and telehealth (N = 2). These topical categories were compared with the 5 areas of unmet research needs in neurotrauma defined by Dasic et al. in 2022, and aligned with service provision, clinical/surgical best practices, prognostication, and follow-up/patients’ engagement [19]. Included articles underwent a level of evidence assessment in accordance with best practices established by Melnyk and Fineout-Overholt (Level I: systematic review or meta-analysis; II: well-designed randomized controlled trial; III: well-designed controlled trial without randomization; IV: well-designed case-control or cohort study; V: systematic review of qualitative or descriptive studies; VI: qualitative or descriptive study; VII: expert opinion or consensus) [25] and are summarized in Table 1. The authors perused the references of these articles and did not find additional items suitable for inclusion in the main review. 

## 3. Results

### 3.1. Epidemiology

Prior Knowledge: While TBI is a long-recognized global epidemic [41], the epidemiology of global pediatric MTBI is under-characterized due to a paucity of reports from developing nations. Prior US studies have identified key rural vs. urban differences, including higher incidence of overall TBI [42], increased TBI severity [43], radiographic intracranial injury [43,44], hospitalized MTBI [45], mortality [41], and reduced access to higher-level trauma centers [20]. Non-US studies have observed similar trends after MTBI/concussion in relative injury severity, and reduced access and outcome [46,47]. Globally, region-specific differences are recognized: for example, rural (vs. urban) MTBIs are less often caused by road traffic accidents in Malaysian and Ethiopian children [21,48], while the reverse is true in US and Canada [44,46].

The current update covers developed nations, where higher-than-previously reported incidences were reported [26,27], as well as developing nations, where multifaceted challenges for epidemiological assessment, injury classification, and triage to care are described for remote regions [28].

Updated Evidence: Langer et al. evaluated the incidence of ED and outpatient concussion visits in Ontario province, Canada, from 2008 to 2016 [26]. Concussion diagnoses were extracted using International Classification of Diseases (ICD) codes. Children accounted for 41–44% of annual visits for concussion care, with the highest incidence in patients aged 0–4 (5400/100,000) [26]. Rurality index correlated with regional concussion incidence, suggesting that rural residence predicted likelihood of MTBI [26]. The updated incidence of MTBI from this study was significantly higher than previous estimates. As the nationalized health care system in Ontario supports healthcare accessibility independent of income, this increased incidence may be more representative of the true incidence for regional pediatric rural MTBI.

In 2013, Feigin et al. conducted a population-based incidence study for TBI in 1 urban and 1 rural district in New Zealand. From 2010 to 2011, the rural MTBI incidence was 1111/100,000 person-years for children aged 0–4 years, and 727/100,000 person-years for those aged 5–14 years [27]. In comparison, the total incidence of rural MTBI across all ages was 758/100,000 person-years. In the Maori ethnic minority, higher MTBI incidence was seen in children aged 0–4 (1346/100,000 person-years) without accounting for differences between rural and urban settings [27]. The authors concluded that the incidence of rural TBI, especially MTBI, in New Zealand is far greater than prior estimates from high-income countries.

Paulino Campos et al. examined TBI patterns in Coari, Brazil, a rural municipality of 85,097 residents [49] served by a single hospital without CT imaging, or neurology or neurosurgery services, and with transportation restricted to air or by sea due to absent road access to other (e.g., more resourced) municipalities [28]. From 2017 to 2019, pediatric TBI accounted for 18% and MTBI accounted for 22% (N = 24) of the 110 total ED visits for TBI [28], although cases for pediatric and MTBI were not specified. The overall transfer rate to higher-level care was 69% (38% for MTBI). The authors noted the low rate of MTBI in their study compared to the literature, and inferred that a high degree of under-reporting was present due to the perceived lack of need for care in a region with inherent barriers to care access. Due to the absence of CT imaging, assessment of TBI severity and classification relied only on GCS and clinical symptoms, and the authors underscore that some of these injuries were likely more severe than “mild”. While the study did not cite specific pediatric MTBI numbers, lack of specialists and ground travel led to further delays (e.g., air flight not available everyday) and suboptimal human and technological monitoring during transport (e.g., boat), which are applicable to properly care for remote rural MTBI.

### 3.2. Care Practices

Prior Knowledge: In the US, rural pediatric TBI incurs greater relative healthcare costs despite having decreased access to inpatient and outpatient TBI specialists, and suffers a multitude of challenges including protracted transport times, geographic restrictions (e.g., inclement weather), and mistriage to non-trauma center [20,22]. In lower- and middle-income countries (LMICs), MTBIs are often unreported and untreated; limited data are captured from urban centers [41] and rural efforts are largely unknown.

The current update examines important trends in healthcare utilization, provider practices and perceptions, and gaps in care and implementation across EDs, primary care, and school-based settings across two large retrospective studies and three focused survey studies in the US and Canada [29,30,31,32,33].

Updated Evidence: Sullivan et al. examined trends in healthcare utilization for pediatric concussions in Ohio state, US, between 2008 and 2016 using a large Medicaid insurance claims database [29]. Of 17,008 patients, 34% were rural and 66% urban. Consistent with the prior literature, increased proportions of rural concussions occurred in patients aged 0–4 and 15–18 years compared to urban. Rates of concussion-related healthcare utilization increased from 2010 to 2013 (2010: rural ~50/10,000, urban 44/10,000; 2013: rural 82/10,000, urban 73/10,000). Following the passage of a statewide concussion law in 2013 that required (1) removal from play after sport-related concussion, (2) health provider clearance before returning to play, and (3) mandatory education for parents and coaches, utilization rates plateaued. For urban patients, initial presentation to ED decreased over time (2008: 50%, 2016: 37%), and to primary care increased over time (2008: 16%, 2016: 48%). Comparatively, for rural patients, initial presentation to ED (2008: 42%, 2016: 51%) and to primary care (2008: 20%, 2016: 38%) increased over time. Follow-up to primary and specialty care steadily increased over time for both urban and rural patients (primary care: ~40% (2008) to ~45% (2016); specialty care: ~20% (2008) to ~35% (2016)). The authors concluded that higher rates of utilization in rural compared to urban areas were commensurate with the known risks for all-cause injury in rural regions. Higher ED utilization in rural (vs. urban) after the 2013 concussion law may be attributable to better integrated ED–primary care systems in rural regions, which may be related to barriers for primary care access.

In 2022, Wittevrongel et al. investigated long-term trends in healthcare utilization for pediatric concussion care in Alberta province, Canada, using the national health services database, which yielded 194,081 ED and physician office visits between 2004 and 2018 (30% from rural/remote) [27]. In total, 52% of visits were initial care, of which 60% occurred in the ED and 40% in physician offices. The overall visit rate increased 2.2-fold over the study period, with a greater increase in physician offices (2.7-fold; 2004: 222/100,000, 2018: 604/100,000) compared to EDs (1.9-fold; 2004: 359/100,000, 2018: 674/100,000). ED utilization for initial care was higher in remote (75% of 2110 visits) and rural regions (76% of 19,654 visits), and lower in urban (60% of 8727 visits) and metropolitan regions (52% of 44,279 visits). Of 36,494 follow-up visits, 23% occurred in the ED and 77% in physician offices, which increased 1.9-fold and 7.1-fold, respectively, over the study period. Greater utilization of the ED for follow-up care occurred in remote (32% of 531 visits) and rural regions (28% of 5128 visits), and less in urban (13% of 2232 visits) and metropolitan regions (21% of 11,686 visits). Overall, initial care increased over the 14-year period, more notably in physician offices. ED utilization for initial and follow-up care remained higher in rural/remote compared to urban regions, which may be linked to financial factors (e.g., lower incomes, higher unemployment rates) and barriers to care (e.g., fewer primary care centers). Increased reporting of “unspecified” head injuries in rural areas may indicate concussion under-reporting relating to the uneven density of accessible care sites [27].

In 2021, Daugherty et al. interviewed 9 US rural primary care providers regarding care practices and perceptions [31]. Providers perceived that concussions were most likely to occur from sports-related and unhelmeted vehicular injuries. Caregivers were more likely to seek initial treatment in the ED rather than primary care, and high rural ED utilization was attributed to the scarcity of rural MTBI specialists. When queried regarding implementation of the 2018 US Centers for Disease Control and Prevention (CDC) guidelines for MTBI management [50], rural providers emphasized that length, accessibility, and usability of proposed clinical tools were most critical to successful adoption, and that tools should be customizable for print and digital formats to improve distribution. Certain proposals, such as the provision of MTBI/concussion-educated staff in schools, were likely impractical due to already constrained human resources in rural settings.

In 2022, Daugherty et al. surveyed 18 US rural ED and primary care providers regarding their care practices for MTBI, including perceived challenges and opportunities for improvement. Providers reported generally adequate ability for diagnosis and care. Consensus challenges included pushback from the community regarding the presence and seriousness of patient injury, lack of specialists available for referral, and logistical (e.g., transportation) and socioeconomic barriers (e.g., income, health insurance). Community education and awareness regarding common MTBI symptoms and sequelae, and the need for medical evaluation after sustaining an injury with suspected MTBI are priority areas for improvement; increased adoption of telemedicine has helped to close part of the gap in access to certain services [32].

In 2021, Pietz et al. examined concussion-related knowledge, confidence, and management experiences through a structured electronic survey of a Washington state, US, cohort of public-school nurses (91% urban, 9% rural) [33]. Across 21 questions, the only statistically significant difference between rural and urban nurses was the correct identification of concussion resolution timeframes (rural: 48% vs. urban: 69% correct). Of the remaining 4 true/false questions on concussion knowledge, correct responses reached 80% for the risk of repetitive concussions, and above 97% for injury mechanisms, loss of consciousness, and resumption of physical activity. Of the 5 questions rating confidence in concussion management, responses ranged from 85% for recognizing that injury has occurred, to 68–72% for providing care/management including return to school education, and to 48% for confidence in management of return to sports/activity. The authors discussed the generally adequate concussion knowledge and confidence in recognition and management across both urban and rural school nurses, as well as the gap in education to both nurses and patients on best practices for return-to-learn and return-to-play. Recommendations were provided for implementing protocols for management, return-to-learn/play, and recognizing the essential role of school nurses in the care for pediatric concussion.

### 3.3. Socioeconomic Factors

Prior Knowledge: In developed nations, large studies have shown that children from racial and ethnic minorities and low-income households experience increased length of hospitalization, mortality, and costs of care after overall TBI compared to socioeconomically advantaged groups [51]. Interactions between SES and pediatric MTBI are more complex, as studies on the effects of SES, symptomatology, and return-to-play after acute concussion in student athletes have been mixed [52,53], despite known associations between SES, history of concussion, and baseline assessments [54,55,56]. Reports on SES disparities in rural settings after MTBI are sparse, even in developed nations.

The current update provides new evidence in SES and income disparities relating to concussion knowledge in student athletes, parental perceptions, and outcomes (post-concussional and mental health symptomatology, objective impairments, health-related quality of life (HRQOL)) in five well-conducted US studies [34,35,36,37,38].

Updated Evidence: In 2020, Chandran et al. examined disparities in concussion knowledge in 541 middle- and high-school student athletes in South Carolina state, US [34], using a structured survey of 18 true/false questions as the primary endpoint. Overall, students scored a median of 10–12 questions correct out of 18. After multivariable correction, urban students (83%) scored higher on concussion than rural (17%) by 1.8 questions. The authors utilized participation in free or reduced-cost school lunch programs as the proxy for lower SES, and participants (64%) scored lower compared to non-participants by 0.5 questions. No differences between urban/rural students were observed for additional questions on perceived seriousness of concussion (median score of 17–18 out of 21) and positive feelings associated with concussion reporting (median score 40–41 out of 45). The authors concluded that rural and low SES may carry risk of decreased concussion knowledge without affecting attitudes or perceptions in students, with implications for context-specific concussion education, and potential benefit for targeted concussion education in schools serving rural and low-income students.

In 2021, Kroshus et al. investigated the influence of household SES on perceived benefits and costs to youth sport participation by surveying 1025 US parents (urban: 88%, rural: 12%; low-income: 34%, middle-income: 35%, high-income: 31%) with at least 1 child aged 5–18 [35]. The survey contained 8 potential benefits and 7 potential costs to sports participation, including risk of concussion, and used a 4-point scale to assign a level of importance or concern. For sports-related concussion risk, a response of “very much a concern” or “somewhat of a concern” was often reported across income levels (low-income: 59%/29%, respectively, middle-income: 38%/35%, high-income: 41%/35%), and 64% of low-income parents viewed keeping their child out of trouble was very important, compared to 40% of high-income parents. The authors stated that differences in the perception of risk between SES groups may be attributable to low-income families’ concerns toward the financial burden of injury (e.g., disability, time for care, decreased access to transportation and healthcare resources).

In 2015, Lin et al. evaluated SES differences in parental understanding of concussions by surveying 214 parents of children receiving orthopedic care at a metropolitan pediatric trauma center and satellite clinics in California state, US [36]. Parents responded to the Concussion Knowledge Index (CKI) and Concussion Attitude Index (CAI) [57]. Overall, the mean CKI was 18.4 on a scale of 0–25 and mean CAI was 63.1 on a scale of 15–75, with higher scores representing greater awareness of concussion features and management practices. Parents earning <30,000 USD/year (7%) had lower mean CKI and CAI (14.9, 54.9) compared to those earning >100,000 USD/year (65%; CKI, CAI: 19.3, 64.7). Parents who did not complete high school (2%) scored lower mean CKI and CAI (12.4, 53.4) compared to parents with a Bachelor’s degree (41%; CKI, CAI: 19.0, 64.6). The authors discussed the importance of understanding the level of parental SES in the context of pediatric concussions, and provision of additional concussion education to parents with lower income and education level.

In 2021, Zonfrillo et al. prospectively examined the relationship between parental education and income on long-term TBI outcomes by studying 123 child–parent pairs across 6 major US hospitals [37]. Children aged 8–18 years with a 30-day history of TBI requiring hospital care were enrolled and assessed at 6, 12, and 24 months post-injury using a concussion symptom inventory (Health Behavior Inventory (HBI)), quality of life survey (Pediatric Quality of Life Assessment (PedsQL)), and executive function/cognitive function measurement (Traumatic Brain Injury-Quality of Life (TBI-QOL). For children with complicated MTBI (GCS 13–15 with intracranial injury on CT), change in health-related quality of life (HRQOL) varied significantly at 12 months post-injury between parental education level of high school (PedsQL: −5.9, from mean baseline: 83.9) vs. parents with Bachelor’s degree or above (PedsQL: +5.1, mean baseline: 85.7) [37]. A similar disparity was observed between children living below 200% of the federal poverty line (FPL; PedsQL: −3.8, mean baseline: 82.3) and those living above 200% of the FPL (PedsQL: +2.0, mean baseline: 85.6). Limited parental education and low household income were also associated with a greater worsening of executive function at 12 months on the TBI-QOL measurement. As outcomes at 6, 12, and 24 months were substantially poorer than baseline for children with the least educated parents, the authors concluded that SES factors may influence functional recovery and HRQOL long after injury, with important implications for post-injury follow-up and rehabilitative care.

In 2019, Connolly et al. retrospectively studied longitudinal mental health symptomatology in 1827 adolescents living in low-income neighborhoods in Chicago, Illinois state, US with a self-reported history of MTBI, and found significantly higher risks of greater-than-baseline aggression, anxiety, depression, attention deficits, and school delinquency compared to their counterparts without MTBI [38]. The study provides initial evidence for the attributable risk of MTBI history with increased psychosocial symptomatology in disadvantaged youths during adolescence, with considerations for MTBI screening as part of behavioral health assessments and therapies.

### 3.4. Telehealth

Prior Knowledge: Adoption of telemedicine platforms in rural healthcare settings has expanded access to specialty neurological care. Virtual triaging and neurologic assessment through telehealth can reduce unnecessary transfers to high-level care facilities and avoid associated costs [20]. Telehealth has been promoted as a solution to address rural–urban disparities in TBI outcomes and provide outpatient assessment and follow-up for medically underserved and geographically remote populations where care access is sparse [20]. In developed nations such as Canada, significant rural and indigenous populations live in healthcare-underserved northern regions and face geographic, structural, and financial barriers to care [58]. As of 2019, preliminary evidence showed telemedicine to be a safe and cost-effective modality to assist in triage and care for well-selected patients with MTBI/concussional symptoms in acute and follow-up care [58]. Well-designed protocols and telemedicine-based network models remain needed.

The current update provides two US studies on implementation of telehealth evaluation (1) for acute TBI in remote and less-resourced regions as part of an integrated transfer protocol to higher-level care [39], and (2) for intensive exercise rehabilitation after MTBI in regions with reduced healthcare access [40].

Updated Evidence: In 2020, Taylor et al. described the experience and lessons learned from instituting a telehealth protocol for remote evaluation of traumatic pediatric injuries by specialists at the state’s only Level I pediatric trauma center in Utah state, US [39]. A TBI-specific sub-protocol outlining clinical and radiographic indications for local admission vs. transfer to the Level I trauma center was implemented at regional hospitals. Seven patients with MTBI received telehealth care and were safely managed without transfer, without requiring readmission. The authors concluded that in the population presenting with low-acuity MTBI to local EDs without onsite specialty care, protocolized telehealth consultation with a higher-level referral center has potential to both reduce unnecessary transfers and provide knowledge on TBI management to non-specialty providers.

In 2021, Chrisman et al. evaluated the feasibility of a novel 6-week telehealth exercise regimen for children with persistent post-concussive symptoms (PPCS) in Washington state, US, developed as a solution for reduced healthcare access during the COVID-19 pandemic [40]. In total, 19 children with a mean of 75 days of PPCS symptoms underwent the Mobile Subthreshold Exercise Program (MSTEP) which involved wearing a wrist fitness tracker (Fitbit), setting exercise heart rate and duration goals, and completing weekly virtual teleconference sessions (Zoom) with the research team. Measurements of concussive symptoms (HBI), fear avoidance (Fear of Pain Questionnaire (FOPQ), and health-related quality of life (Peds-QL) were assessed. Subjects were highly compliant with study procedures and showed steady improvements in symptomatology across all measurements over the course of 6 weeks (HBI: −11 points; FOPQ: −22 points; PedsQL: +15 points). These findings demonstrated not only the feasibility but also the potential efficacy of telehealth-mediated interventions during concussion recovery, which can increase the accessibility of well-designed rehabilitative services.

## 4. Discussion

The global incidence of pediatric MTBI/concussion is increasing relative to the total number of TBIs, according to the recent Global Neurotrauma Outcomes Study [59]. Consequently, its impact on morbidity, disability, and public health burden continues to rise. Rurality is known to confer increased risk of high-velocity injuries, geographic and financial barriers to high-acuity and specialty care access, decreased availability of human resources, and increased relative costs after MTBI. The knowledge update from our current work provides several crucial findings across four topical domains (Figure 2): considerable increases in injury incidence compared to before, with implications for targeted MTBI/concussion awareness education and injury reporting; differential care utilization over time and contrasts between rural and urban regions; provider self-reports on their ability to diagnose and manage MTBI/concussion; education priorities and counseling strategies; socioeconomic risk factors for symptomatology, outcome, and barriers to care; and novel telehealth protocols for acute and outpatient care. The dearth of modern reports from developing nations and LMICs on rural and underserved pediatric MTBI remains a significant limitation toward actionable plans for improvement, and constitutes another knowledge gap in care requiring awareness and care deficiency to be addressed.

### 4.1. Epidemiology

Our review found that in developed nations, the recent incidence of concussion in all youths, and notably, 0–4-year-olds are considerably higher than historical reports. While historical rates generally do not exceed 300/100,000 in developed nations [1,5,6,60,61], Sullivan et al. reported overall rates of 440–500/100,000 in 2010, which increased to 730–820/100,000 in 2013 [29]. Modern rural healthcare utilization rates consistently exceed urban rates by 1.1- to 1.3-fold [29,30]. More concerningly, in 2020 Langer et al. found that the rate of annual visits in a predominantly rural Canadian province was 1153/100,000 overall, and 5400/100,000 for the high-risk group of ages 0–4 [26]. While the study was inclusive of EDs and outpatient settings and a subset are follow-up visits, this finding nevertheless implicates the urgent need to better understand potential underlying causes, such as the increased awareness of injury and need for care, and decreased financial barriers to care (e.g., insurance status) within a nationalized healthcare system, but also potentially drastic under-reporting even in developed nations. These focused questions should be addressed by large retrospective and prospective efforts.

In developing nations and LMICs, generally with less resourced healthcare organizations and complex payment systems, under-reporting and decreased awareness are likely amplified [41]. Indeed, Paulino Campos et al. showed that MTBI accounted for only 22% of total ED TBI visits to the single local hospital in the remote Amazonas due to lack of perceived medical need for MTBI to present to care. Even so, 38% of MTBI patients were transferred to higher levels of care due to the absence of diagnostic CT imaging, and neurosurgery and neurology services at their hospital [28]. Transfer to another hospital from rural/remote regions accrues additional risks to protracted time to care and patient safety [20,28], as vehicles are constrained by craft availability and inclement weather, mis-triage may occur, and specialized personnel and/or medical equipment may be absent.

Understanding the modern epidemiology of MTBI/concussion is requisite to acquiring insight into the changing landscape of injury, and the first step toward assessment of care practices, risk factors, process improvements, and potential technological integrations. The scarcity of recent studies in rural and underserved regions in LMICs emphasizes the critical need to support clinicians and researchers from these regions.

### 4.2. Care Practices

Recent evidence from the US and Canada demonstrated several salient trends in healthcare utilization for pediatric MTBI. Rural patients often presented to EDs over primary care for initial care [29,30]. In the US, Sullivan et al. reported from primarily low-income households that rural patients continued to utilize the ED more so than primary care over time (2008: 42% (ED) vs. 20% (primary care); 2016: 51% vs. 38%). In contrast, urban patients shifted toward presenting to primary care rather than EDs (2008: 50% (ED) vs. 16% (primary care); 2016: 37% vs. 48%). The proportion presenting to follow-up care were similar across ruralities, and increased over time (primary care: 40% to 45%; specialty care: 20% to 35%) [29]. In Canada, Wittevrongel et al. reported comparable results using national health services data over 14 years. For initial care, ED utilization was highest in remote/rural (75–76%) and lowest in urban (60%) and metropolitan regions (52%); similarly for follow-up care, ED utilization was highest in remote/rural (28–32%) and lowest in metropolitan (21%) and urban regions (13%). Both studies showed increased utilization over time for initial care (US: 1.7-fold over 8 years; Canada: 2.2-fold over 14 years) [29,30]. As alluded to in the Section 4.1, the observed increase in incidence may be attributable to improved data recording and coding methods (e.g., electronic health records) and improved injury awareness in more educated regions (e.g., urban/metropolitan) to offset under-reporting, rather than an absolute increase in the volume of injuries. As shown by Sullivan et al., rates of care utilization plateaued following the 2013 concussion law, and urban ED utilization decreased while rural utilization slightly increased [29]. Mandatory clearance by a healthcare provider prior to return to play likely shifted a majority of standardized MTBI/concussion care to outpatient settings, and mandatory education to caregivers and coaches likely improved injury awareness, as evidenced by the increase in specialty care over time.

Rural reliance on ED care is related to its accessibility, as the distribution of outpatient offices is likely sparse across a large region and outpatient care is limited to daytime hours and transportation barriers. This was confirmed through rural US primary care provider interviews by Daugherty et al. [31]. Certain US rural EDs are integrated into referral networks with primary care providers, incentivizing caregivers to present to the ED [29]. Within developed nations, ED utilization was significantly higher in Canada compared to the US across ruralities, implicating a role of the differential costs of care between a nationalized healthcare system (Canada), versus a mixed system with coexisting government and privately funded health insurance coverage (US). In 2020, the average per-person cost of a US ED visit was USD 420 for Medicaid-insured and USD 560 for privately insured patients, compared to 158 Canadian Dollars (CAD) for Canadian patients (1 CAD converts to approximately 0.75 USD in 2023) [62,63]. Whereas all Canadian citizens have identical insurance coverage, greater proportions of rural US children are covered by low-income/government-funded insurance compared to urban [64,65,66].

Our review identified actionable provider perceptions in rural settings. Daugherty et al. reported that management guidelines require concision, applicability, and customizability to be successfully adopted by resource-constrained settings [31]. The US CDC HEADS UP initiative website has implemented expert feedback and serves as a source for providing MTBI/concussion fact sheets, acute and post-acute assessments, prevention and referral strategies, concussion policies, and other important resources, and is customizable by the viewer’s role/occupation across print and digital formats [67]. Rural ED, primary care, and school nursing providers reported confidence in diagnosis and management of MTBI/concussion [31,32,33] and cited geographic and socioeconomic barriers to care. School nurses reported highest concern about their lack of formal training in return-to-activity counseling [33]; bolstering this type of training will empower nurse providers to better care for their students and serve as an invested source to disseminate MTBI/concussion education.

In a 2020 US multicenter study of 600 concussion youths with outcomes data, 40% remained symptomatic at 3 months and 20% at 6 months post-injury [68]; US treatment costs for pediatric MTBI are estimated at USD 1.3 billion overall and USD 1868 per patient within 1 year post-injury [10]. As such, there is a growing impetus to develop, implement and refine health policies in order to create solutions for rural healthcare delivery, increase the number of trained personnel for primary and specialty care, and provide urgent financial and logistical resources to tackle these barriers, which require coordinated efforts amongst rural leadership, governmental institutions, funding agencies, and industry partners. Providing updated education to patients, providers, schools, and community leaders regarding the seriousness of MTBI/concussion injury, symptomatology, and potential for lasting deficits is equally important, not only to expedite care but moreover, to foster community investment in injury prevention, safety, and paradigm change. Our review identifies this as an addressable challenge across care and school settings.

Task shifting and task sharing, defined as delegation of clinical care to non-neurosurgeons, is ongoing in hospital systems where neurosurgeons are scarce [69]. Based on surveys from both high-income countries and LMICs, there was consensus that these represented practical approaches to optimizing the workforce to tackle the burden of neurotrauma across adult and pediatric services. Perspectives differed across neurosurgical care providers regarding the circumstances for which task shifting and sharing should be implemented to enhance care in human-resource-scarce regions while minimizing disruptions to traditional training. They also called for considerations in regulation and certification processes for both in-person and telemedicine models [69], which can be applied to pediatric MTBI/concussion care practices in rural and underserved regions.

### 4.3. Socioeconomic Factors

Our review revealed several key attributes of lower SES for children and parents in knowledge, perception, and outcome in rural MTBI/concussion. Chandran et al. found that low SES was associated with decreased concussion knowledge in rural US athletes, without notable differences in perception of concussion seriousness or reporting when compared to urban athletes [34]. Accordingly, in the setting of limited resources, education to disadvantaged rural children should focus on recognition of concussion symptoms and prevention strategies to improve patient awareness and expedite closure of the knowledge gap. Connolly et al. found that US adolescents in low-income neighborhoods with history of MTBI had elevated mental health and attentional symptoms and academic delinquency compared to those without MTBI [38], underscoring the importance of integrating MTBI screening, provider training, and referral networks into behavioral health assessments for socioeconomically disadvantaged youths.

Low education and income were associated with decreased concussion knowledge and awareness in US parents; moreover, low education (no high school degree, vs. completed Bachelor’s degree) was associated with greater decreases in knowledge and awareness compared to low income (<30,000, vs. >100,000 USD) [36]. Zonfrillo et al. provided objective data in children with complicated MTBIs, where low parental education and income was associated with poorer executive function at 1 year post-injury, and HRQOL at 1 year worsened from 30 days post-injury for patients with parents who completed high school, while HRQOL improved for children whose parents had completed Bachelor’s degrees [37]. These recent data indicate that in addition to rurality, children with decreased parental education incur increased risk for timely injury recognition and indicated care. High-level data on parental education and SES can be collected with a few short questions and should be systematically integrated into acute and follow-up MTBI/concussion evaluations in rural and under-resourced settings. High-risk cohorts should alert providers to closely counsel, follow-up, and provide educational materials to both children and parents to mitigate long-term morbidity and improve outcomes at a population level.

### 4.4. Telehealth

The development of comprehensive telehealth protocols in recent years reflects recent advances in its accessibility and delivery for rural populations. Taylor et al. showed that telehealth can be integrated with MTBI-specific triage and assessment at local emergency and acute care centers with the assistance of a well-resourced center capable of providing requisite expertise and accepting transfers for higher-level care if deemed necessary [39]. As such, large hospitals should work with local/regional hospitals to develop emergency consultation and referral networks for trauma care, and telehealth best practices should be pursued in resource-poor regions. In parallel, Chrisman et al. showed the efficacy of telehealth as the primary treatment modality delivered during an exclusively remote intensive outpatient rehabilitation program over the course of 6 weeks in patients with PPCS to reduce symptomatology and improve HRQOL [40]. Feedback included high patient/caregiver compliance, and appreciation for the structured and guided exercise program, support from research staff, and tracking progress from mobile fitness trackers. Augmentation of internet access is critical to enable the potential of telehealth in care improvement and cost savings. Furthermore, telehealth is a modality in which interactive and updated education can be provided to local providers and patients.

Implementation of telehealth protocols for rural/remote regions in both developed nations and LMICs should be prioritized for MTBI/concussion care in order to improve geographic and financial access to care, optimize remote decision-making, reduce burden to constrained care systems, and improve outcomes. In Canada, development of regional MTBI/concussion protocols that incorporate risk stratification, injury etiology, prior assessments, and telemedicine access to identify patients who can be managed remotely through telehealth are recently underway [58].

### 4.5. Limitations

We recognize several limitations. Our literature review was restricted to articles indexed in the National Library of Medicine PubMed library to ensure adequate analysis by the study authors. While we designed our search criteria to be widely inclusive of keywords and MeSH headings/subheadings, and no article returned from our search was excluded based on English language, utilization of international literature databases and study personnel fluent in analyses of non-English reports may yield additional topical evidence. The majority of articles returned from our search originated from countries with English as the primary language (e.g., US, Canada, New Zealand), resulting in a selection bias for papers from developed countries with comparatively robust healthcare and technological infrastructure. The limited geographic diversity and relative scarcity of data from regions designated as rural and underserved in LMICs limit the generalizability of observations on emerging modalities, e.g., applications of telehealth, which were drawn from studies conducted in well-resourced care settings. We relied on our comprehensive search terms to guide us on the selection of reports specific to rural and underserved regions, and despite our best efforts, it is possible that certain reports from these regions may have been missed if they were not designated with a focus on rural or underserved MTBI/concussion populations. Our review aimed to provide a focused knowledge update on relevant modern evidence for pediatric MTBI/concussion in rural and underserved settings, and therefore we did not re-summarize topical papers included in prior reviews. We assessed the level of evidence for included studies [25], and did not systematically review and grade the breadth of historical evidence, which was out of scope of the current review. More granular assessment of quality was also not performed, as the aim of the current work was to provide a summary of the evidence, rather than producing more stringent practice recommendations [70].

## 5. Conclusions

Pediatric MTBI/concussion patients in rural/underserved regions experience increased risks of injury, geographic and financial barriers to care, and poorer outcomes. Globally, under-reporting of injuries has hindered our epidemiological understanding. Decreased awareness of the seriousness of MTBI/concussion in the treated population is a major challenge for rural providers. Ongoing MTBI education should be provided to rural caregivers, schools, and low-income populations to improve community understanding of symptomatology/disability, indicated care, return-to-play considerations, and prevention strategies. MTBI management guidelines should be concise, applicable, and customizable to their target under-resourced settings to facilitate adoption. Telehealth advancements as part of integrated acute triage and consultation protocols, and outpatient care and rehabilitation programs, have potential to decrease unnecessary transfers, transportation delays, and costs. Modern research on MTBI/concussion risk factors and care practices from LMICs is greatly needed.

## Figures and Tables

**Figure 1 jcm-12-03309-f001:**
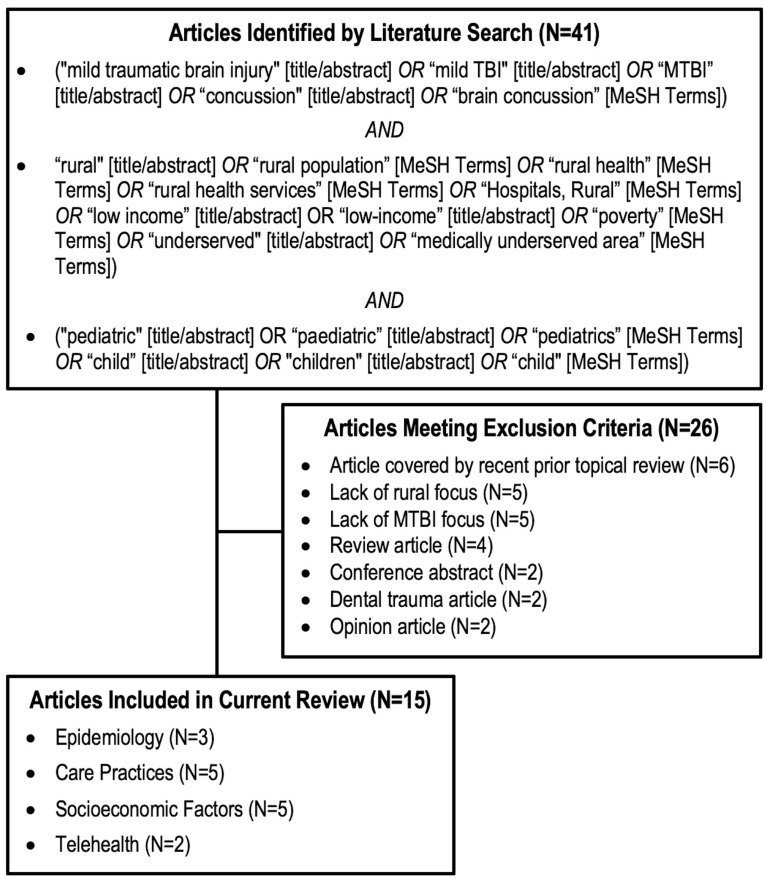
Flowchart of included studies.

**Figure 2 jcm-12-03309-f002:**
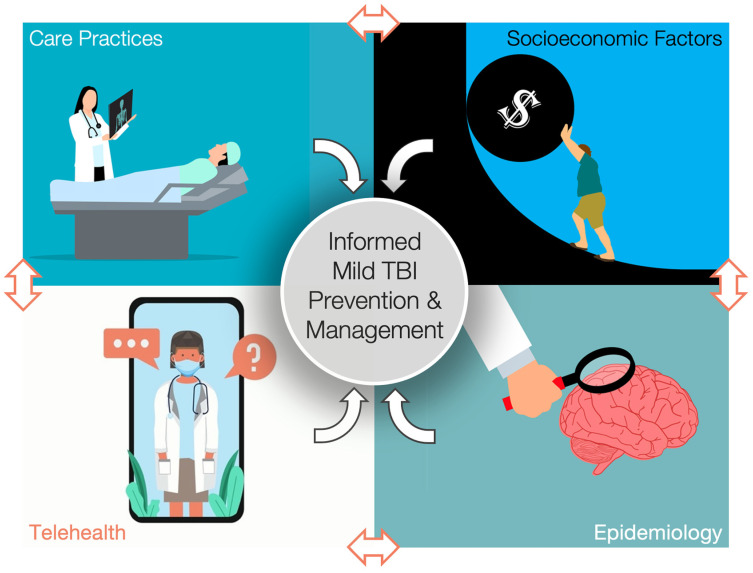
Topical categories of contemporary findings on injury prevention, gaps in care, and management strategies in rural and underserved pediatric mild traumatic brain injury (TBI) research. Development, implementation, and refinement of targeted best practices, as well as processes and pathways for reducing disparities, across these 4 domains should be prioritized in pediatric mild TBI research and clinical care.

**Table 1 jcm-12-03309-t001:** Included Rural Pediatric MTBI/Concussion Studies by Topic.

Author (Year)	Study Type; Country	Sample Size	Cohort Characteristics	Description	Measurements	Results	Conclusions	Level of Evidence
**Epidemiology**								
Langer et al. (2020) [26]	Retrospective epidemiological study; Canada	1,330,336 concussion patients; Jan 2008–Dec 2016.	Mean age: 30 y. M/F: 55/45%. Adult/Pediatric: 58/42%.	Patients with concussion extracted from a national registry using ICD codes.	Demographics; temporal and geographic characteristics; level of initial care.	Concussion incidence was highest in children aged 0–4 y (5400/100,000) and was positively associated with rurality (r = 0.67).	For MTBI, young children in rural areas constitute a vulnerable subpopulation in Canada.	IV
Feigin et al. (2013) [27]	Prospective epidemiological study; New Zealand	1369 TBI cases (MTBI: N = 1298); Mar 2010–Feb 2011.	MTBI median age: 21 y. M/F: 62/38%. Rural/urban: 26/74%.	TBI registry in rural city (Waikato; N = 43,956) and urban city (Hamilton; N = 129,249).	Demographics; injury etiology; level of initial care.	Overall rural MTBI incidence: 758/100,000. MTBI incidence for rural children aged 0–4 y: 1111/100,000.	The incidence of rural MTBI in New Zealand exceeds prior estimates from high-income countries.	III
Paulino Campos et al. (2021) [28]	Retrospective epidemiological study; Brazil	110 cases of rural TBI (MTBI N = 24; Pediatric N = 20); Jan 2017–Oct 2019.	MTBI: M/F 70/30%	TBI patients at the only hospital in a rural Brazilian municipality (85,097 residents).	Demographics; injury etiology, transfers, inpatient length of stay.	No CT imaging, neurology/neurosurgery services at hospital, and 69% of TBI patients (38% of MTBI) required transfer to higher level of care. No medical ground transport in remote Coari, Brazil—only air and boat.	Low MTBI incidence may be linked to deferral of care in healthcare-limited settings. Lack of services and transportation may lead to misclassified TBI severity and care delays.	VI
**Care Practices**								
Sullivan et al. (2021) [29]	Retrospective cohort study; US	17,008 pediatric concussion patients; Jan 2008–Dec 2016.	M/F: 64/36%. Rural/urban: 66/34%.	Medicaid claims database study of 330,000 low-income children in Ohio.	Level of initial care; type of follow-up care.	ED utilization for initial concussion care decreased in urban (2008: 50%, 2016: 37%) and increased in rural settings over time (2008: 42%, 2016: 51%). Utilization of ED for initial care plateaued after passage of 2013 concussion law. Follow-up to primary and specialty care increased over time.	In Ohio, greater rural compared to urban ED utilization may be an effect of better integrated ED–primary care systems in rural regions.	III
Wittevrongel et al. (2022) [30]	Retrospective cohort study; Canada	194,081 unique concussion encounters; Apr 2004–Mar 2018.	Median age: 13 y. M/F: 64/36%.	Pediatric concussion records from Alberta’s provincial healthcare system registry.	SES; level of initial care; locality/rurality.	Higher rates of ED utilization for initial and follow-up concussion care observed in remote (75/32%) and rural regions (76/28%) compared to urban (60/13%) and metropolitan regions (52/21%).	Primary care visits are increasingly utilized for initial care in urban settings. ED continues to be utilized for care in rural and remote settings.	III
Daugherty et al. (2021) [31]	Descriptive study; US	9 pediatric rural health care providers.	US primary, urgent, and ED care providers.	Structured interview of rural providers treating pediatric MTBI.	Providers’ subjective experiences, attitudes, and beliefs.	Providers noted patient preference for ED initial concussion care. Requests for imaging and limited financial resources impede management.	Rural providers would benefit from increased financial, educational, and personnel resources to implement care and prevention guidelines.	VI
Daugherty et al. (2022) [32]	Descriptive study; US	18 pediatric rural health care providers and social workers.	Medicine, mental health, ED providers; social workers.	Structured, interview-based survey of rural healthcare providers for children with MTBI.	Providers’ subjective experiences, attitudes, and beliefs.	Consensus challenges: community pushback regarding the seriousness of patient injury, lack of specialists, logistical (e.g., transportation) and SES barriers (e.g., income, health insurance).	Reduction of SES barriers to care access will improve MTBI management. Community education on the seriousness of MTBI and its sequelae is greatly needed.	VI
Pietz et al. (2021) [33]	Descriptive study; US	315 school nurses; Nov 2016–May 2017.	School location urban/non-urban: 91/9%.	Structured, internet-based survey of urban and rural school nurses.	Knowledge of concussion and evidence-based management guidelines.	Rural and urban school nurses have comparable knowledge, differing only knowledge of recovery timeframe (rural: 48% vs. urban: 69% correct).	Continuing education for school nurses can ensure up-to-date practice of pediatric MTBI care.	VI
**Socioeconomic Factors**								
Chandran et al. (2020) [34]	Observational study; US	541 youth student athletes.	M/F: 63/37%. Rural/urban: 17/83%.	Structured survey of concussion knowledge and attitudes among student athletes.	Demographics; concussion knowledge score (18-point scale); concussion attitudes.	In middle and high school athletes, disparities in concussion knowledge were observed for urban vs. rural (urban: +1.8 correct responses) and SES status (low SES: −0.5 correct responses).	Rural and low SES students had lower concussion-related knowledge than urban and high SES students, with implications for targeted concussion education.	VI
Kroshus et al. (2021) [35]	Observational study; US	1025 parents of youth sports participants.	Parent mean age: 43. M/F: 44/56%.	Structured, internet-based survey of parents with child aged 5–18 y.	Household demographic; attitudes toward youth sports.	Concussion is “very much a concern” or “somewhat of a concern” for parents of all SES (low-income: 59/29%, middle: 38/35%, high-income: 41/35%).	Differences in perception of concussion risk between SES groups may be influenced by low-income families’ concerns toward the financial burden of injury.	VI
Lin et al. (2015) [36]	Observational study; US	214 parents of children with sports-related concussions.	Parent M/F: 27/73%.	Structured, survey-based assessment of parents.	Concussion Knowledge Index (CKI) and Concussion Attitude Index (CAI) scores.	Disparities in mean CKI/CAI scores were observed in household SES (<30,000 USD/year: 14.9/54.9; >100,000 USD/year: 19.3/64.7) and education (less than high school: 12.4/53.4; Bachelor’s degree: 19.0/64.6).	Parental concussion education provided to parents with limited secondary education or low-income may improve concussion awareness and detection in youth sports.	VI
Zonfrillo et al. (2021) [37]	Prospective cohort study; US	170 children with TBI (MTBI N = 123); Mar 2013–Feb 2015.	Mean age: 13.2 y; M/F: 72%/28%; Rural/urban: 15/85%.	Prospective study of children with 30-day history of TBI who sought care at 1 of 6 US hospitals.	Demographics; injury etiology; symptomatology (HBI); executive function (TBI-QOL); HRQOL (PedsQL)	Lower QOL at 12 months associated with low parental education (PedsQL; high school: −5.9 from baseline 83.9; college: +5.1 from 85.7) and lower household income (<200% Federal Poverty Level (FPL): −3.8 from 82.3; >200% FPL: +2.0 from 85.6).	Socioeconomic factors can predict long-term recovery of functional status in pediatric MTBI patients.	III
Connolly et al. (2019) [38]	Prospective cohort study; US	1827 low-income children aged 10–18 y.	Median age: 11 y; M/F: 50/50%.	Longitudinal study of psychosocial symptoms in adolescents with low SES +/− history of MTBI.	Demographics; SES, injury; psychosocial symptoms.	Adolescents with self-reported MTBI history are significantly more likely to experience aggression, depression, anxiety, attention-deficit, and academic delinquency.	Additional supportive care for management of psychosocial symptoms may benefit disadvantaged adolescent MTBI patients.	III
**Telemedicine**								
Taylor et al. (2021) [39]	Retrospective cohort study; US.	72 pediatric trauma patients; Jan 2019–Feb 2020.	Age range: 7 m–15 y	Patients presenting to a local facility affiliated with Utah’s Level I pediatric trauma center.	Clinical characteristics; outcomes.	Evaluation of a novel telehealth protocol. In 72 pediatric trauma patients, 8 telehealth consults (7 MTBI) occurred; 0% transferred to higher level of care, 0% readmissions.	A pediatric TBI telehealth protocol integrated into triage and referral care systems may improve resource utilization without negatively affecting quality of care.	VI
Chrisman et al. (2021) [40]	Prospective observational study; US.	19 children with PPCS recruited from a concussion clinic; 2018–2019.	Mean age: 14.3 y; M/F: 21/79%.	Prospective study of a novel telehealth-delivered exercise intervention for children with PPCS.	Demographics; symptomatology (HBI); HRQOL (PedsQL); Fear of Pain; at 3 and 6 weeks in program.	Patients showed increased HRQOL (PedsQL: +15), decreased symptomatology (HBI: −11 points), and decreased fear of pain (−22 points). High compliance rate and positive feedback regarding participation.	Telehealth may enable clinically and cost-effective post-concussion care and rehabilitative services, including intensive and long-term follow-up care.	VI

Caption: Detailed descriptions of included studies. Level of evidence was assessed in accordance with best practices established by Melnyk and Fineout-Overholt (details in Section 2) [25]. CAI = Concussion Attitude Index; CKI = Concussion Knowledge Index; ED = emergency department; CT = computed tomography; F = female; FPL = federal poverty level; HBI = Health Behavior Inventory; HRQOL = health-related quality of life; M = male; m = month; MTBI = mild traumatic brain injury; PedsQL = Pediatric Quality of Life Assessment; PPCS = persistent post-concussive symptoms; QOL = quality of life; SES = socioeconomic status; TBI = traumatic brain injury; US = United States; USD = United States dollar; y = year.

## Data Availability

This work was a literature review of articles available in the National Library of Medicine PubMed library.

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
