# Peer review of "Update on Pediatric Mild Traumatic Brain Injury in Rural and Underserved Regions: A Global Perspective"

_jcm, 2023, doi:10.3390/jcm12093309_

Round 1

Reviewer 1 Report

The study defines the correlation between developmental and psychosocial factors in determining outcome/prognosis in mild pediatric TBI. The study examines and consolidates previous literature across different categories to support their claim. The authors have established strategies/clinical implementations to address healthcare deficits in addressing mild TBI. The study also provides solutions in improving care delivery and community awareness. The study's only limitation is the number of literature/articles that has been used to systematically put together this review. Also, the authors can improve on stating whether this literature collection that they used (15 articles) are reliable and reputed sources to support their claim. The manuscript is well written with a clear conclusion. Overall, a well presented literature survey review that fills in the knowledge gap in mTBI.

Author Response

Response to Reviewers

Reviewer 1

The study defines the correlation between developmental and psychosocial factors in determining outcome/prognosis in mild pediatric TBI. The study examines and consolidates previous literature across different categories to support their claim. The authors have established strategies/clinical implementations to address healthcare deficits in addressing mild TBI. The study also provides solutions in improving care delivery and community awareness. The study's only limitation is the number of literature/articles that has been used to systematically put together this review. Also, the authors can improve on stating whether this literature collection that they used (15 articles) are reliable and reputed sources to support their claim. The manuscript is well written with a clear conclusion. Overall, a well presented literature survey review that fills in the knowledge gap in mTBI.

Author Response: We thank the Reviewer for their thorough evaluation and constructive comments. We agree on the limitations of our review, which was structured as a narrative review of recent topical evidence in rural and underserved pediatric MTBI/concussion, and does not constitute a systematic review. This was previously discussed in our Limitations section (Lines 584-586): Our review aimed to provide a focused knowledge update on relevant modern evidence for pediatric MTBI/concussion in rural and underserved settings, and therefore we did not re-summarize topical papers included in prior reviews.”

We appreciate the Reviewer’s suggestion on whether the included articles “are reliable and reputed sources”. To better address this, we have improved the Methods section (Lines 109-110): Included articles underwent level of evidence assessment in accordance with best practices established by Melnyk and Fineout-Overholt (Level I: Systematic review or meta-analysis; II: Well-designed randomized controlled trial; III: well-designed controlled trial without randomization; IV: well-designed case-control or cohort study; V: Systematic review of qualitative or descriptive studies; VI: Qualitative or descriptive study; VII: Expert opinion or consensus) [25] and are summarized in Table 1.” We have assessed all included studies and provided their level of evidence in Table 1.

We have added to the Limitations (Lines 587-589): We assessed the level of evidence for included studies [25], and did not systematically review and grade the breadth of historical evidence, which was out of scope of the current review.”

Reviewer 2 Report

Unlike previous studies which focused on mTBI in rural areas of developed nations, Yue and colleagues systematically reviewed current gaps in the prevention and management of acute and post-acute paediatric mTBI from a global perspective. This study addressed a critical knowledge gap needed to inform the development of targeted solutions.

The manuscript is sufficiently detailed and well written. The methods used to identify and review the relevant papers are appropriate. I have no changes to suggest.

Author Response

Response to Reviewers

Reviewer 2

Unlike previous studies which focused on mTBI in rural areas of developed nations, Yue and colleagues systematically reviewed current gaps in the prevention and management of acute and post-acute paediatric mTBI from a global perspective. This study addressed a critical knowledge gap needed to inform the development of targeted solutions.

The manuscript is sufficiently detailed and well written. The methods used to identify and review the relevant papers are appropriate. I have no changes to suggest.

Author Response: We thank the Reviewer for their careful review of our manuscript and support of our findings.

Reviewer 3 Report

This manuscript by Dr. Yue and colleagues reviews pediatric care of Mild TBI (MTBI) patients in rural and underserved populations globally. TBI, and especially MTBI, inflicts significant healthcare, societal, and economic burdens globally and particularly in the pediatric population who are at increased risk for detrimental TBI sequalae. Thus, it is imperative to better understand the current epidemiology, risk factors, and care practices associated with TBI and that differ among various communities to improve management. As the authors note, a major challenge in assessing TBI care in the literature is the underreporting of patients and/or reduced awareness among patients and even healthcare providers, especially in rural areas and developing nations. To this end, the authors identified 15 articles, which were not included in prior reviews, to discuss current updates to the epidemiology, care practices, socioeconomic factors, and telehealth practices involved in global MTBI care in developing/rural populations. From their review, the authors highlight several important topics: (1) Firstly, their review of current epidemiology indicates that more recent papers have shown a higher reported incidence of TBI in the pediatric population than previous reports, both in developed nations as well as rural areas. The authors insight into potential reasons for the increase in incidence, such as improved awareness and reporting over time among other causes, highlights important avenues for future research. Of which, the authors note the lack of recent studies in low-mid-income countries and heed the need for such investigations. (2) Secondly, analysis of care utilization across multiple studies showed that patients in rural areas are much more likely to utilize the ED compared to those living in metropolitan or urban areas. The authors make a very interesting point that the ED utilization rate is higher in Canada, whom have nationalized healthcare, compared to the US, where healthcare is mixed between government-funded and private health insurance. This raises an intriguing point of how health insurance differences across communities affects TBI care. (3) The authors also note several socioeconomic factors, such as education, income, and school resources/knowledge of school nurses, as distinguishing factors in rural versus urban TBI care. These are definitely important factors to consider and address through various policies and initiatives to overcome these socioeconomic barriers. (4) Lastly, the authors highlight how telehealth can be used as a modality to improve TBI care in rural regions and developing countries. This is an astute and paramount aspect to consider, especially with the current rise in telehealth and technologies supporting telehealth post-COVID-19 pandemic.

Overall, the review is written very clearly, organized, and data in the flowchart and table are well-presented. The work provides a comprehensive assessment of TBI/MTBI care in underserved areas with important points for follow-up studies to investigate. I enthusiastically support publication of this manuscript in Journal of Clinical Medicine. Some minor grammatical suggestions are detailed below:

-       Line 20 of abstract: should read “Incidences are higher for…, and ages 0-4 years and are increasing over time.”

-       Line 169: replace “and” after “transport times” with a comma

-       Line 366: no comma is needed after “procedures”

-       Line 385: no comma is needed after “improvement”

-       Line 462: no comma is needed after “MTBI/concussion”

-       Line 493: no comma is needed after “networks”

-       Line 505: no comma is needed after “questions”

-       Line 533: no comma is needed after “telehealth”

-       Line 542: insert comma after “US”

-       One potential suggestion for an additional summary schematic figure of the various factors assessed in the study may be beneficial for the reader, although not necessary.

Author Response

Response to Reviewers

Reviewer 3

This manuscript by Dr. Yue and colleagues reviews pediatric care of Mild TBI (MTBI) patients in rural and underserved populations globally. TBI, and especially MTBI, inflicts significant healthcare, societal, and economic burdens globally and particularly in the pediatric population who are at increased risk for detrimental TBI sequalae. Thus, it is imperative to better understand the current epidemiology, risk factors, and care practices associated with TBI and that differ among various communities to improve management. As the authors note, a major challenge in assessing TBI care in the literature is the underreporting of patients and/or reduced awareness among patients and even healthcare providers, especially in rural areas and developing nations. To this end, the authors identified 15 articles, which were not included in prior reviews, to discuss current updates to the epidemiology, care practices, socioeconomic factors, and telehealth practices involved in global MTBI care in developing/rural populations. From their review, the authors highlight several important topics: (1) Firstly, their review of current epidemiology indicates that more recent papers have shown a higher reported incidence of TBI in the pediatric population than previous reports, both in developed nations as well as rural areas. The authors insight into potential reasons for the increase in incidence, such as improved awareness and reporting over time among other causes, highlights important avenues for future research. Of which, the authors note the lack of recent studies in low-mid-income countries and heed the need for such investigations. (2) Secondly, analysis of care utilization across multiple studies showed that patients in rural areas are much more likely to utilize the ED compared to those living in metropolitan or urban areas. The authors make a very interesting point that the ED utilization rate is higher in Canada, whom have nationalized healthcare, compared to the US, where healthcare is mixed between government-funded and private health insurance. This raises an intriguing point of how health insurance differences across communities affects TBI care. (3) The authors also note several socioeconomic factors, such as education, income, and school resources/knowledge of school nurses, as distinguishing factors in rural versus urban TBI care. These are definitely important factors to consider and address through various policies and initiatives to overcome these socioeconomic barriers. (4) Lastly, the authors highlight how telehealth can be used as a modality to improve TBI care in rural regions and developing countries. This is an astute and paramount aspect to consider, especially with the current rise in telehealth and technologies supporting telehealth post-COVID-19 pandemic.

Overall, the review is written very clearly, organized, and data in the flowchart and table are well-presented. The work provides a comprehensive assessment of TBI/MTBI care in underserved areas with important points for follow-up studies to investigate. I enthusiastically support publication of this manuscript in Journal of Clinical Medicine. Some minor grammatical suggestions are detailed below:

-       Line 20 of abstract: should read “Incidences are higher for…, and ages 0-4 years and are increasing over time.”

-       Line 169: replace “and” after “transport times” with a comma

-       Line 366: no comma is needed after “procedures”

-       Line 385: no comma is needed after “improvement”

-       Line 462: no comma is needed after “MTBI/concussion”

-       Line 493: no comma is needed after “networks”

-       Line 505: no comma is needed after “questions”

-       Line 533: no comma is needed after “telehealth”

-       Line 542: insert comma after “US”

-       One potential suggestion for an additional summary schematic figure of the various factors assessed in the study may be beneficial for the reader, although not necessary.

Author Response: We thank the Reviewer for their detailed and thorough review of our manuscript and support of our findings. We have responded to the Reviewer comments in detail below.

-       Line 20 of abstract: should read “Incidences are higher for…, and ages 0-4 years and are increasing over time.”

Author Response: We have revised this sentence accordingly.

-       Line 169: replace “and” after “transport times” with a comma

Author Response: We have revised this sentence accordingly.

-       Line 366: no comma is needed after “procedures”

Author Response: We have revised this sentence accordingly.

-       Line 385: no comma is needed after “improvement”

Author Response: We have revised the sentence in accordance to the Editor’s comments, and the comma after the word “improvement” now improves the flow of the sentence.

-       Line 462: no comma is needed after “MTBI/concussion”

Author Response: We have revised this sentence accordingly.

-       Line 493: no comma is needed after “networks”

Author Response: We have revised this sentence accordingly.

-       Line 505: no comma is needed after “questions”

Author Response: We have revised this sentence accordingly.

-       Line 533: no comma is needed after “telehealth”

Author Response: We have revised this sentence accordingly.

-       Line 542: insert comma after “US”

Author Response: We have revised this sentence accordingly.

-       One potential suggestion for an additional summary schematic figure of the various factors assessed in the study may be beneficial for the reader, although not necessary.

Author Response: We have added a new Figure 2 (Lines 405-410) to illustrate the 4 domains of contemporary findings in rural/underserved pediatric mild TBI research summarized in our review.
